# Adaptation of the Endolithic Biome in Antarctic Volcanic Rocks

**DOI:** 10.3390/ijms241813824

**Published:** 2023-09-07

**Authors:** Andrea Hidalgo-Arias, Víctor Muñoz-Hisado, Pilar Valles, Adelina Geyer, Eva Garcia-Lopez, Cristina Cid

**Affiliations:** 1Center for Astrobiology (CAB), CSIC-INTA, Torrejón de Ardoz, 28850 Madrid, Spain; andhid01@ucm.es (A.H.-A.); vmhisado@cab.inta-csic.es (V.M.-H.); evanit3@hotmail.com (E.G.-L.); 2Materials and Structures Department, National Institute of Aerospace Technology (INTA), Torrejón de Ardoz, 28850 Madrid, Spain; vallesgp@inta.es; 3Geosciences Barcelona (GEO3BCN), CSIC, Lluís Solé Sabarís s/n, 08028 Barcelona, Spain; ageyer@geo3bcn.csic.es

**Keywords:** endolithic microorganisms, Deception Island, extremophiles, volcanic rock, 16S/18S rRNA sequencing, bioinformatics, astrobiology

## Abstract

Endolithic microorganisms, ranging from microeukaryotes to bacteria and archaea, live within the cracks and crevices of rocks. Deception Island in Antarctica constitutes an extreme environment in which endoliths face environmental threats such as intense cold, lack of light in winter, high solar radiation in summer, and heat emitted as the result of volcanic eruptions. In addition, the endolithic biome is considered the harshest one on Earth, since it suffers added threats such as dryness or lack of nutrients. Even so, samples from this hostile environment, collected at various points throughout the island, hosted diverse and numerous microorganisms such as bacteria, fungi, diatoms, ciliates, flagellates and unicellular algae. These endoliths were first identified by Scanning Electron Microscopy (SEM). To understand the molecular mechanisms of adaptation of these endoliths to their environment, genomics techniques were used, and prokaryotic and eukaryotic microorganisms were identified by metabarcoding, sequencing the V3–V4 and V4–V5 regions of the 16S and 18S rRNA genes, respectively. Subsequently, the sequences were analyzed by bioinformatic methods that allow their metabolism to be deduced from the taxonomy. The results obtained concluded that some of these microorganisms have activated the biosynthesis routes of pigments such as prodigiosin or flavonoids. These adaptation studies also revealed that microorganisms defend themselves against environmental toxins by activating metabolic pathways for the degradation of compounds such as ethylbenzene, xylene and dioxins and for the biosynthesis of antioxidant molecules such as glutathione. Finally, these Antarctic endolithic microorganisms are of great interest in astrobiology since endolithic settings are environmentally analogous to the primitive Earth or the surfaces of extraterrestrial bodies.

## 1. Introduction

The endolithic biome consists of microbes living in tiny pores and cracks in rocks. Endoliths have colonized a large range of environments, from the Earth’s surface to the deep biosphere, and several millimeters under the subsurface. This biome can be considered the harshest one on Earth, due to a near-complete absence of sunlight, oxygen, and most nutrients [1]. But, in turn, the rocks offer a microenvironment that protects from intense solar radiation and desiccation, as well as sources of nutrients, moisture, and substrates derived from minerals [2,3]. So far, cryptoendolithic organisms such as lichens in the Dry Valleys of Antarctica have been described [4]. But most endoliths include members of bacteria, archaea and microeukaryotes. The adaptation of microorganisms to this extreme biome is of great interest in various fields of research, such as diverse as ecology, biotechnology and astrobiology. For instance, microorganisms living beneath the ocean floor have been suggested to play a role in the carbon and sulfur cycles, and global warming [5]. Other endolithic microorganisms may contribute to the bioremediation of contaminated soils [6]. In addition, these organisms may be responsible for the biomineralization of economically important mines [7]. *Cyanobacteria* are the most widespread endolithic organisms [1]. They constitute the oldest vestige of life that has been recorded in the Paleoproterozoic in volcanic environments [8,9,10]. Endolithic fungi have also been described in volcanic rocks [11], as well as bacteria that feed on organic matter or minerals and can establish consortia and symbiotic relationships [12,13].

Among volcanic environments, Deception Island (South Shetland Islands) is one of the most active volcanoes in Antarctica with a record of more than 20 explosive eruptions in the last two centuries [14,15]. It consists of a composite volcano located at the southwestern end of the Bransfield Strait [16]. The normal magnetic polarity of all Deception Island’s exposed rocks indicates that these are younger than 0.75 Ma [17], and K-Ar data [18] point to 0.2 Ma. The morphology of the island is strongly marked by a central c. 8.5 × 10 km collapse caldera (Figure 1A). The caldera-forming event, with 30–60 km^3^ of ejected magma, has been dated at c. 8300 years, according to paleomagnetic data [19] and 3980  ±  125 calibrated years Before Present (cal yr BP) based on tephrochronology, sedimentological studies and 14C dating [20]. Previous to the caldera collapse, volcanic activity led to the formation of multiple shoaling seamounts and a main subaerial volcanic edifice [15] (Figure 1BI). During the caldera-forming event, a several-tens-of- meters-thick sequence of pyroclastic density current deposits was emplaced [15,21,22] (Figure 1BII). It consists of a sequence of massive dense pyroclastic density current deposits, several tens of meters thick, and forms an almost continuous outcrop that encircles most of the island. The post-caldera phase, which includes the more recent historical eruptions (1829–1970), consists of at least 70 dispersed eruptive vents distributed along the structural borders of the caldera and the interior of Port Foster, the sea-flooded part of the caldera depression (Figure 1A). These have mainly consisted of hydromagmatic eruptions with variable intensity and explosivity degree, mostly depending on the amount and provenance of water that interacted with the rising magma [23]. Historic eruptions have been concentrated in eruptive clusters (e.g., 1818–1828, 1906–1912, 1967–1970) with several temporally closely spaced events, separated by decades of quiescence (e.g., 1912–1967). The recorded historical volcanic activity (1967, 1969 and 1970), and the episodes of unrest which occurred in 1992, 1999 and 2014–2015, undoubtedly categorize Deception Island as a very active volcano.

Endolithic organisms inside the volcanic rocks generated in these eruptions could be long-term descendants of organisms that colonized the soils before or after eruptions. They could live inside the rocks or be the remains of dead organisms or fossils that have been preserved in the lava. Our study aims at identifying how endolithic microbial communities are composed in the extreme environment of Deception Island volcanic rocks and elucidating how they adapt to the environment. To analyze the endolithic niches, we applied contamination-minimized sampling procedures and extracted DNA from the inner parts of the rocks. Several environmental parameters such as rock type and chemical composition of rocks, along with microscopic observation, were analyzed together to explain patterns of microbial diversity. Finally, from the taxonomic assignments of the microbial populations, metabolic pathways were inferred by genomic and bioinformatic methods.

## 2. Results and Discussion

A summary of the overall experimental strategy is represented in Appendix A.

### 2.1. Mineral Phase and Chemical Composition of Volcanic Rocks

The identification of the minerals by X-ray diffraction (XRD) showed that the rocks were mainly composed of anorthite, augite, labradorite and albite (Figure 2, Appendix A). Examples of spectra from samples of volcanic rocks analyzed by XRD are shown in Appendix A.

The samples were also analyzed by mass ICP to quantify the chemical elements contained in the samples (Appendix A). It was observed that the chemical composition of the two groups of samples (pyroclastic density current deposits versus loose pyroclastic lapilli-size deposits) was different (Appendix A). According to the ANOVA analysis, all the samples had a different chemical composition as a whole (*** *p* < 0.0001). In addition, when comparing the composition of the two types of samples separately, the Student’s *t*-test showed that both groups also were significantly different (### *p* < 0.0001).

### 2.2. Identification of Microorganisms by SEM

When the samples were microscopically observed by SEM, it was possible to perceive elements that visually looked like microorganisms (Figure 3). In previous reports, the presence of diatoms in lake sediments from Deception Island has been described [20]. In our samples, in addition to diatom frustules (Figure 3B,C,J), cocci bacteria (Figure 3D,F) and bacilli bacteria (Figure 3K) were detected. Many microorganisms found in rock samples were fungi, in both filamentary (Figure 3A,D,E) and globular (Figure 3I) forms. In some samples such as DIVOL_23, crysophytes were observed (Figure 3L). Pollen grains with diverse morphology (Figure 3H) and even wood remains (Figure 3G) were also distinguished.

### 2.3. Endolith Viability

In order to verify if the microorganisms observed inside the rocks were alive or were only remains of dead microorganisms, several viability experiments were carried out, as described in materials and methods.

The results of these experiments showed that cell growth was observed in the cultures of the inner part of the rocks, which means that the microorganisms inside the rocks were alive. Fragments from the outside of the rocks were also cut away and cultured, but those cultures did not grow, as the outer part had been irradiated with UVC light (Appendix A).

Part of the identified DNA sequences may belong to viable but nonculturable environmental microorganisms (VBNC). Therefore, experiments on endolith viability quantification are not representative of the total microbial population. Even so, the living microorganisms should be those that were mostly identified by sequencing and those that were observed intact in the SEM images. In addition, it could be concluded that the DIVOL_4A and DIVOL_23 samples presented lower viability, perhaps because these samples came from less porous rocks and the microorganisms had more difficulty accessing the interior.

### 2.4. Microbial Community Composition

According to the metabarcoding results, a total of 259,163 bacterial ASVs with an average length of 566 base pairs, corresponding to 868 genera and spanning 24 phyla, were obtained (Figure 4A; Appendix A). Only 215 (0.1%) sequences could not be identified. The most abundant bacterial phyla were Proteobacteria (52.3%), followed by *Actinobacteriota* (23.2%) (Appendix A). The relative abundance of bacteria was higher in the pyroclastic rock–loose lapilli samples than in the pyroclastic density current deposit samples (Appendix A). It is possible that lapilli, being a more porous material, allows microorganisms to enter the interior of the rock more easily. In some samples (DIPV_36_S3, DIVOL_12A, DIVOL_4A), the representatives of the phylum *Bacillota* were very abundant. And it is interesting to highlight the richness of *Bacteroidota* in the samples from density current deposits. The most abundant genera were *Ralstonia*, *Gaiella* and *Polaromonas*. These three bacterial genera had previously been described in polar regions [26] and glacier ice archives [27]. The great abundance of the genus *Ralstonia* in the samples was outstanding (Appendix A). *Ralstonia* is a Gram-negative, aerobic, and facultative chemolithoautotrophic bacilli [28], with a great capacity to deal with hostile living conditions, such as extreme pH levels or oligotrophic environments [29]. It should be noted that the abundance gradient of this genus in the samples coincides directly with the number of metabolic processes related to quorum sensing and the formation of biofilms, DIPV36_S3 being the sample with the highest relative abundance of the genus *Ralstonia* (54.7%) and with a greater abundance of enzymes related to these processes. It is known that *Ralstonia* is capable of forming biofilms, creating a hydrated matrix of proteins, polysaccharides and nucleic acids, constituting a unique communication and defense system [30]; therefore, it is likely that the key to survival and dominance of *Ralstonia* in this habitat is this formation of biofilms. However, cyanobacteria, which are traditional endolithic microorganisms in other environments, were only 2% of the bacterial population in these samples (Appendix A). DIVOL_23 was the sample most exposed to the sea, since it was collected on the outer coast of the island (Figure 1A). The remaining samples were collected in areas located inside the volcanic caldera, and their microbial populations have less interaction with the ocean. This sample is the one with the greatest diversity of microorganisms. For example, the bacterium, *Himenobacter*, has been described as an indicator of population exchange between sea and land in the Arctic Archipelago Svalbard [31].

Among the endolithic microorganisms, bacteria adapted to the extreme temperatures of Deception Island were found. Temperatures range from the intense winter cold that reaches −30 °C to the heat coming from the volcanic caldera (temperatures of 70 °C have been recorded around the steam vents and geothermal waters). Among the psychrophiles, for example, the genera *Psychroglaciecola* and *Psychrobacter* were identified. Among the thermophiles were representatives of the class *Thermoleophilia*, the order *Thermomicrobiales* and the genera *Thermomonas* and *Thermaerobacter*.

Contrary to prokaryotes, eukaryotic sequencing yielded a large number of unassigned sequences (37.5%) (Figure 4B; Appendix A). A total of 33,296 eukaryotic ASVs with an average length of 712 base pairs were obtained. Surprisingly, in one of the samples, DIPV_36_S3, DNA corresponding to eukaryotes could not be detected (Figure 4; Appendix A). In the SEM images of this sample, eukaryotes could not be observed, either; only bacteria were detected. (Figure 3F). The most abundant eukaryotic groups were *Opisthokonta* (39.9%), which were mostly fungi, and *Archaeplastida* (22.7%) (Appendix A). Among *Archaeplastida*, the most abundant algae were *Chlorophyta*, *Raphidonema* and *Trebouxia*. The genus *Ancylonema*, characteristic of pink snow, was very abundant in the sample DIVOL_4A [32]. Only in rocks of the pyroclastic density current deposit type were representatives of the SAR phylum identified, especially *Stramenopiles*, *Ciliophora* and *Cercozoa*. Among *Stramenopiles*, unicellular nanoflagellates as *Bicosoecida* were found in the DIVOL_23 sample. They are free-living cells, with no chloroplasts, and are encased in a lorica [33]. This microorganism is characteristic of plankton from the deep ocean to surface waters. Its presence on the rocks must be caused by the proximity of this sample to the shoreline.

Regarding the low abundance of microeukaryotes, it is possible that after the most recent eruptions of 1969 and 1970, almost all eukaryotic diversity disappeared, and some of the results obtained in this study are either dead microorganisms or fossil DNA. So, the actual microeukaryotes would have secondarily reached the island by other means, such as animals, humans or environmental factors, colonizing these volcanic rocks. This may be the case for pollen grains observed by microscopy, since on Deception Island, there is practically no native vegetation.

### 2.5. Microbial Distribution According to Rock Type

To determine if there was any kind of relationship between the type of volcanic rocks studied and the microbial populations, several multivariate statistical analyses were performed [34]. First, several PCA analyses were performed with the prokaryotic and eukaryotic samples separately. In both analyses, it was observed that the microbial populations were segregated into two groups, which correspondeded to the two rock types analyzed: pyroclastic density current deposits and loose pyroclastic lapilli-size deposits (Figure 5). The first axis of the PCA explained 78.9% of the total variation for bacterial phyla (Figure 5A and Appendix A analysis no. 1), and 95.7% for bacterial genera (Figure 5B and Appendix A analysis no. 2). The first axis was mainly driven by the abundance of *Burkholderia*, *Desulfitibacter* and *Ralstonia*. The percentages for eukaryotes were 63.3% and 72.3% for phyla and genera, respectively (Figure 5C,D; Appendix A analyses 5, 6). The distribution of the populations into two groups was according to the type of sample. *Deinococcota* was the only phylum of bacteria located in both sample types.

### 2.6. Microbial Distribution According to Chemical Composition

Since it had already been verified that the two types of volcanic rocks studied were chemically different (Appendix A), an attempt was made to verify if chemical composition could influence the different distributions of microbial populations. To carry out this test, the CCA using the concentrations of the dissolved ions from rocks was used. The results of this analysis demonstrated that ion concentrations were more closely related to the distribution of the prokaryotic phyla (78.4%) than to the eukaryotic phyla (61.7%) (Figure 6; Appendix A).

It was observed that some groups of bacteria appeared closely related to a certain ion. For example, the genus *Ferruginibacter* was related to iron, and the genera *Desulfitibacter*, *Desulfofundulus* and *Desulfotomaculum* were related to sulfur concentrations (Figure 6B). The genus *Hymenobacter* looked related to chlorine, possibly due to the proximity of these samples to the sea. The transport of this bacterial genus between coastal and marine environments has been previously mentioned [31].

Both in the SEM images and the DNA sequencing, the presence of wood was detected (Figure 3G). This could be the origin of the organic matter used by fungi. The presence of ancient forests in this region of Antarctica is known; although no traces of forests have been found on Deception Island, they have been described on nearby Livingston Island [35]. However, this vegetable matter, if it is still usable by fungi, must be much more recent than the fossils described, or could even come from historic wood structures [36].

### 2.7. Elucidation of Microbial Adaptation to the Environment Inferred through Taxonomy

The software PICRUSt2 predicts the functional potential of a microbial community based on marker gene sequencing profiles. In this research, the predictions of the metabolic pathways from the taxonomy showed that the adaptations of the microbial populations differed from sample to sample (Figure 7; Appendix A). Some metabolic routes were, to a greater or lesser degree, common to all samples, such as the fatty acid synthesis pathways (Figure 7). Another metabolic process common to nearly all samples was oxidative phosphorylation (Appendix A). Other metabolic pathways such as the pentose phosphate pathway are shown in red in the diagrams of samples DIVOL_12A and NB1_S2, and in blue in the diagrams of samples DIVOL_4A and DIVOL_23, meaning 1% and 5% abundance, respectively. It is noteworthy that the two samples that correspond to the pyroclastic density current deposit group (DIVOL_4A and DIVOL_23) were the ones with the highest microbial diversity and with the highest metabolic activity (Appendix A).

In some samples, the metabolic pathways corresponding to the TCA cycle and amino acid metabolism were very activated for bacteria. And unexpectedly, in the sample DIVOL_23, the routes of biodegradation of compounds such as ethylbenzene, dioxins and xylene were active (Appendix A). The degradation of these compounds by microorganisms has been studied in depth for its usefulness in the decontamination of water and soil around petrochemical production factories [37,38].

Surprisingly, in one of the samples, DIPV_36_S3, no eukaryotes were identified. And the bacterial population in this sample had a very simple metabolism (Appendix A). The metabolic pathways and processes of the microbial consortium may be more complex when the populations are more diverse, i.e., when the microbial population consists of both prokaryotes and eukaryotes. The two samples with the highest eukaryotic diversity were the samples with the most complex metabolism. In samples NB1_S1, DIPV36_S1 and DIPV36_S2 (Figure 3A,D,E), the growth of fungal hyphae was observed by SEM. These hyphae must have originated from outside the rocks and internalized into them through small pores. The fungal growth, in turn, facilitates the weathering of the rocks and the opening of internal channels [39]. These fungi have a heterotrophic metabolism and degrade organic matter. For example, in sample DIVOL_4A (Figure 3G), wood remains were observed. The carbon obtained from their degradation and its transport through the hyphae into the rocks would facilitate the growth of other microorganisms such as heterotrophic bacteria.

Some autotrophs can synthesize their organic compounds by using gas or dissolved nutrients from water droplets moving through fractured rock [40]. Others may incorporate inorganic compounds from rock substrates, perhaps by excreting acids to dissolve the rock [41]. Various chemoautotrophic bacteria that metabolize iron and sulfur, such as *Ferruginibacter*, *Acidiphilium*, *Desulfotomaculum*, *Desulfitibacter* and *Desulfofundulus* were identified in the metabarcoding analyses (Appendix A). In the samples DIVOL_4A and DIVOL_23, the presence of the *Paracoccus* bacterium, which is capable of oxidizing reduced sulfur compounds, was detected. Appendix A shows the activation of the pathway corresponding to sulfur metabolism in the sample DIVOL_23. These types of iron and sulfur metabolizing microorganisms had already been detected in the ice of the Deception Island glaciers [26,42].

Many endoliths are primary phototrophic producers and eroders of substrata spanning both prokaryotic (cyanobacteria) and eukaryotic algae [1,43]. *Chrysophyte* cysts were observed in the SEM images, in the sample DIVOL_23 (Figure 3L). However, they could not be seen in their vegetative form. DNA from various *Chrysophyceae* was detected in samples DIVOL_4A and DIVOL_23, from pyroclastic density current deposits rocks. It had already been reported that chrysophytes can live in a wide variety of metal-rich environments such as volcanic rocks [44].

Fragments of diatom skeletons could be observed in the SEM images (Figure 3B,J). This raises the possibility that they are the remains of frustules from before the last volcanic eruptions to the formation of the rocks. Even so, DNA corresponding to diatoms of the genus *Fistulifera* was detected in the DIVOL_4A sample. Diatom DNA was not detected in the rest of the samples.

Some microorganisms synthesize a red pyrrole-containing pigment called prodigiosin. These molecules are produced as a secondary metabolite and play a role in energy transfer, as porphyrins, chlorophylls and phycobilins do [45]. Flavonoid biosynthesis has also been reported in bacteria and fungi. Both synthesis pathways, prodigiosin and flavonoid biosynthesis, were activated in microorganisms from various samples, especially prokaryotes (Appendix A). The presence of pigments such as chlorophylls is associated with photosynthesis in the endolithic microorganisms that inhabit the superficial areas of rocks, accessible to sunlight. In addition, endolithic microorganisms present strategies against stress that include the synthesis of several types of colored molecules. Among them, compounds such as carotenoids and melanin act as photoprotective-screening and photoprotective-quenching pigments. They had already been described in cyanobacteria, algae and fungi from other non-Antarctic endolithic biomes [1].

Finally, it is noteworthy that the metabolism of glutathione was detected, especially in bacteria, and only in eukaryotes from the DIVOL_23 sample (Appendix A). Glutathione, the most important intracellular redox buffer, is found mainly in eukaryotes and Gram-negative bacteria. However, some other prokaryotes, which are characterized by the absence of glutathione, produce other low-molecular-weight thiols which possibly fulfill the same functions as glutathione [46].

Endolithic microorganisms are the main inhabitants in both warm [47] and polar [48] desert areas, and in high mountains [49]. The existence of endolithic microorganisms in marine and fluvial sediments has also been extensively reported [50]. All of these endolithic biomes present common adaptations, as they are dark, anaerobic ecosystems with slow metabolic rates [50]. A summary of the main adaptive mechanisms of microorganisms to endolithic environments is represented in Appendix A. Compared to the abovementioned endolithic biomes, microorganisms in volcanic Antarctic environments reveal a special adaptation. These biomes are inhabited by encased microorganisms that are highly dependent on the composition of their host rocks. For example, the microorganisms of marine and fluvial sediments are not stressed by the lack of water and, therefore, their synthesis of molecules to avoid desiccation is not common. These submerged biomes contain a large population of heterotrophs, which is much smaller in the volcanic environments of Antarctica. Thus, the biome studied in this research is much more similar to biomes in dry desert environments, both cold and warm. Adaptation strategies specific to volcanic Antarctic environments include autotrophic metabolism using sulfur-based compounds as energy sources, iron and magnesium oxidation, and the synthesis of specific pigments. In contrast, some strategies typical of endolithic environments of aqueous marine or riverine environments do not appear in the Antarctic volcanic endolithic biome. Other detailed strategies for adaptation to the environment cannot be known in detail with the techniques used in this research. It would be necessary to use other analytical techniques (i.e., proteomics) to find out if the cellular machinery for the synthesis of specific molecules is present in these microorganisms.

### 2.8. Implications for Astrobiology

The habitability of Antarctic volcanic rocks in Deception Island may alter our understanding of life [1]. Endolithic settings are environments analogous to the primitive Earth or the surface of extraterrestrial sceneries such as Venus, Mars, Europa, Enceladus and Titan [51]. The discovery of living microorganisms in these harsh environments, or even only the finding of microbial fossils or endolithic biosignatures, is of great interest in astrobiology because endolithic microorganisms can live within rocks, thereby escaping damaging UV rays and other threats to which they may be subjected outside Earth’s atmosphere [52].

The demonstration of the existence of living microorganisms inside the Antarctic volcanic rocks would allow the present research to be expanded by subjecting these extremophile populations to extraterrestrial conditions such as those that can be studied in simulation chambers, since these are highly resistant microorganisms that could survive such conditions [53].

## 3. Materials and Methods

### 3.1. Sample Collection

Studied volcanic rock samples were collected during the 2017–2018 and 2018–2019 Spanish Antarctic campaigns in the framework of the POSVOLDEC and VOLCLIMA projects. Different locations distributed over the entire island were sampled (Figure 1A and Figure 2A). Samples were wrapped in sterile plastic bags and stored until use. All 11 samples correspond to either pyroclastic density current deposits (partially consolidated) or loose pyroclastic lapilli-size deposits (Figure 2A) (Table 1). Pictures of some examples of the Antarctic volcanic rocks are shown in Appendix A. Despite no precise age being available for the individual rock samples, they belong to eruptions occurring after the caldera collapse (i.e., within the last 8300 years if paleomagnetic data is considered). Rigorous efforts were made at all stages to control for contamination, including sampling and laboratory work.

### 3.2. Identification of Mineral Phase and Chemical Composition of Volcanic Rock Samples

The identification of the mineral phase was performed by XRD. The identification of the pulverized minerals was carried out with the Diffract.Eva v.7 software (Bruker, Bremen, Germany) and the PDF2 mineral database (https://www.icdd.com/, accessed on 26 July 2023) [54].

Concentrations of ions in volcanic rock samples were analyzed by inductively coupled plasma-mass spectrometry (ICP-MS) on a Perkin Elmer ELAN9000 ICP-MS quadrupole spectrometer, as in [42]. Values > 10 ppm were considered for the statistical tests.

### 3.3. Scanning Electron Microscopy (SEM)

Microorganisms on volcanic rock samples were observed by field emission scanning electron microscopy (FE-SEM), using a ThermoScientific (Boston, MA, USA) APREO C-LV microscope, with an AZTEC ORFORD energy dispersive X-ray (EDX) microanalysis system. The volcanic rock samples were fragmented in half to expose the interior, and the inner core was observed. Then, the samples were coated with 4 nm of chromium using a sputtering Leica EM ACE 600. The images were obtained at 10 kV [55].

### 3.4. Processing of Samples and Analysis of the Viability of Endolytic Microorganisms

In order to check whether the endolytic microorganisms were alive or not, the outer areas of each rock were sterilized, and individual cultures were prepared. The surfaces of volcanic rock samples were irradiated with UVC light (germicidal lamp (VL-6.MC, Vilber Lourmat, France) at an intensity of 400 μW/cm^2^ for 45 min, changing their position every 15 min to sterilize the surface [56].

Each rock was individually cut into separate fragments from the external and internal zones (named outer part and inner part, respectively, Appendix A), and crushed in an agate mortar in sterile conditions. Each sample was aerobically incubated in R2A medium for 3 days with agitation at 150 rpm, at 30° in the dark. Growth was monitored by optical density at 600 nm (Appendix A). Three biological replicates were grown. All procedures were performed by using bleach-sterilized work areas, a UV-irradiated laminar flow hood, ethanol-sterilized tools and sterilized gloves.

### 3.5. DNA Extraction and Sequencing

DNA was sequenced from 5 samples belonging to either pyroclastic density current deposits (partially consolidated) or loose pyroclastic lapilli-size deposits (Table 1). Approximately 500 mg of each sample was used for DNA extraction. Samples NB1_S2, DIVOL_12A and DIPV_36_S3 were representative of the first group, and samples DIVOL_4A and DIVOL_23 were representative of the second group. DNA was extracted from one aliquot from each volcanic rock (only the inner fraction) using FastDNA™ SPIN Kit for Soil (QIAGEN, Hilden, Germany). DNA quantification was carried out with the commercial Quant-iT™ dsDNA HS Assay kit (Invitrogen™, Waltham, MA, USA). Its concentration was determined with the Qubit™ fluorimeter (Invitrogen™). The diversity of uncultured microeukaryotes and bacteria was assessed by Illumina MiSeq 16S and 18S rRNA gene amplicon sequencing [57]. The amplification and sequencing of the V3-V4 regions of the 16S rRNA gene (forward sequence CCTACGGGNGGCWGCAG; reverse sequence GACTACHVGGGTATCTAATC) were performed to identify bacteria. Microeukaryotes were identified by amplification and sequencing of the V4-V5 regions of the 18S rRNA gene (forward sequence GCCAGCAVCYGCGGTAAY; reverse sequence CCGTCAATTHCTTYAART) [58].

### 3.6. Metabarcoding Data Processing

Two bioinformatic analyses were carried out: one for the identification of bacteria (16S rRNA), and the other for microeukaryotes (18S rRNA). The QIIME2 (v2022.8) software [59] was used to process the reads obtained from the sequencing. The primers were removed with the CUTADAPT tool (via q2-cutadapt) [60], and the sequences were processed with DADA2 (via q2-dada2) [61] to obtain the representative sequences. DADA2 parameters were (--p-trunc-len-f 272 \; --p-trunc-len-r 199) and the SILVA 138-99 database for bacteria and p0 (--p-trunc-len-f 0 \; --p-trunc-len-r 0) and the SILVA 132-99 database for microeukaryotes.

To decide which database was the most suitable for identifying the microorganisms contained in the samples, the ASVs obtained with DADA2 were compared against three different databases (SILVA 132-97, SILVA 132-99 and SILVA 138-99). This procedure was repeated for each of the analyses. The criteria for the selection of parameters in the analysis of the sequences were: (i) the quality of the sequences, (ii) the number of representative ASVs obtained and their length, and (iii) the presence or absence of chimeras. An attempt was made to make use of the most up-to-date database that gave consistent results. After observing the results of the bioinformatic analysis for the 16S rRNA region, it was observed that using the pX parameters, a greater number of sequences, a higher mean and a lower standard deviation were obtained with respect to the length of the ASVs. Regarding the taxonomy, although the SILVA 132-97 and 132-99 databases gave very similar results, the SILVA 138-99 database was used, since it was the most up-to-date. With these parameters, the sequences corresponding to mitochondria and chloroplasts were very low in abundance, and were removed by hand. Using these criteria, no archaea were detected. For the analysis of microeukaryotes (18S rRNA), according to the same selection criteria, it was decided to use the p0 parameters, since, although more ASVs were obtained with the X parameters, all ASVs were classified as “Unassigned” or “Ambiguous taxa”.

### 3.7. Prediction of Metabolic Pathways

The prediction of metagenome functions was developed with the PICRUSt2 software using the amplicon sequence variants (ASVs) [62]. This software creates predictions about the metabolic pathways taking place in the samples by assigning a taxon to each different sequence, taking into account its abundance. So, it gives an overall picture of the biochemical processes occurring in each environment. The analysis was performed on the data concerning bacterial and eukaryotic diversity. The ASVs corresponding to the most representative genera obtained in the analysis of the 16S and 18S rRNA genes were isolated. A matrix was obtained with the abundances of the different enzymes in each sample analyzed, and a KO (KEGG Orthology) number was assigned to each enzyme [63]. The KO number is a unique identifier for each protein that allows its classification according to its biological function, and its comparison among them. Five analyses were performed (one for each sample) for bacteria and five for eukaryotes, observing the 1% most abundant enzymes and the 5% most abundant enzymes in each sample. The KO numbers respective to these percentages were collected, and ten metabolic maps were constructed with the KEGG Mapper Color tool. Thus, the most abundant metabolic pathways in the microbial community of each sample were captured.

## 4. Conclusions

-A great diversity of endolithic microorganisms was found colonizing the volcanic rocks of the Antarctic Deception Island. These rocks provide microorganisms with a source of nutrients, a water reservoir, and a microclimate that protects them from low temperatures and solar radiation.-Much more bacterial than eukaryotic diversity was found in these volcanic rocks, since they better withstand extreme environments. Prokaryotes utilize survival mechanisms and different nutrition strategies, since both chemolithoautotrophic and photoautotrophic bacteria were present in these populations.-The interiors of the rocks host living organisms whose mechanisms of molecular adaptation can be elucidated thanks to the omic and bioinformatic sciences. The results of these analyses have shown fundamental and common metabolic pathways such as the biosynthesis and degradation of fatty acids, the metabolism of amino acids, the TCA cycle, and the metabolism of sulfur and nitrogen.-Some of these endolithic microorganisms also synthesize pigments such as prodigiosin and flavonoids, which act as photoprotective-screening and photoprotective-quenching pigments. Other microorganisms found in these samples produce enzymes capable of degrading contaminants, such as ethylbenzene, dioxins and xylene, to defend themselves against environmental toxins.-This extreme ecosystem is a good terrestrial analogue of extraterrestrial environments. The demonstration of the existence of living microorganisms inside the Antarctic volcanic rocks broadens the scenarios for the search for life on other planets from an astrobiological perspective.

## Figures and Tables

**Figure 1 ijms-24-13824-f001:**
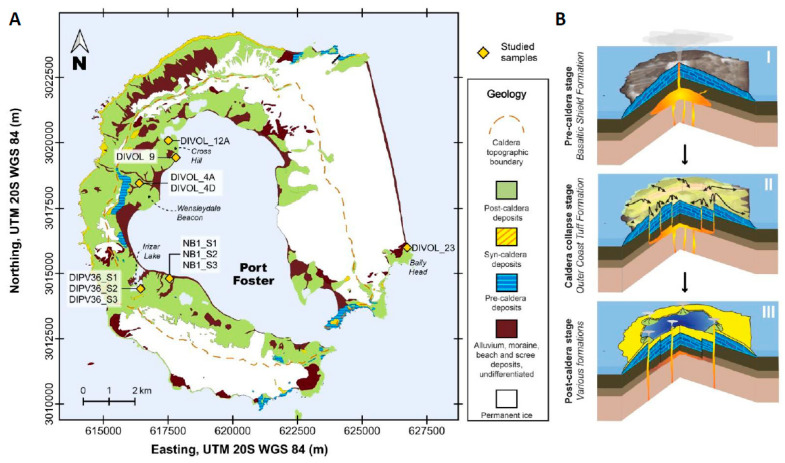
(**A**) Simplified geological map of Deception Island. Map showing the location of the studied samples [21] (modified from [15]). Data obtained from Spatial Data Infrastructure for Deception Island SIMAC [24,25]. (**B**) Volcanic evolution. The three cartoons present the volcanic evolution of the island starting with the composite volcano formation, followed by the caldera collapse, and finishing with the post-caldera eruptive episodes (modified from [21]).

**Figure 2 ijms-24-13824-f002:**
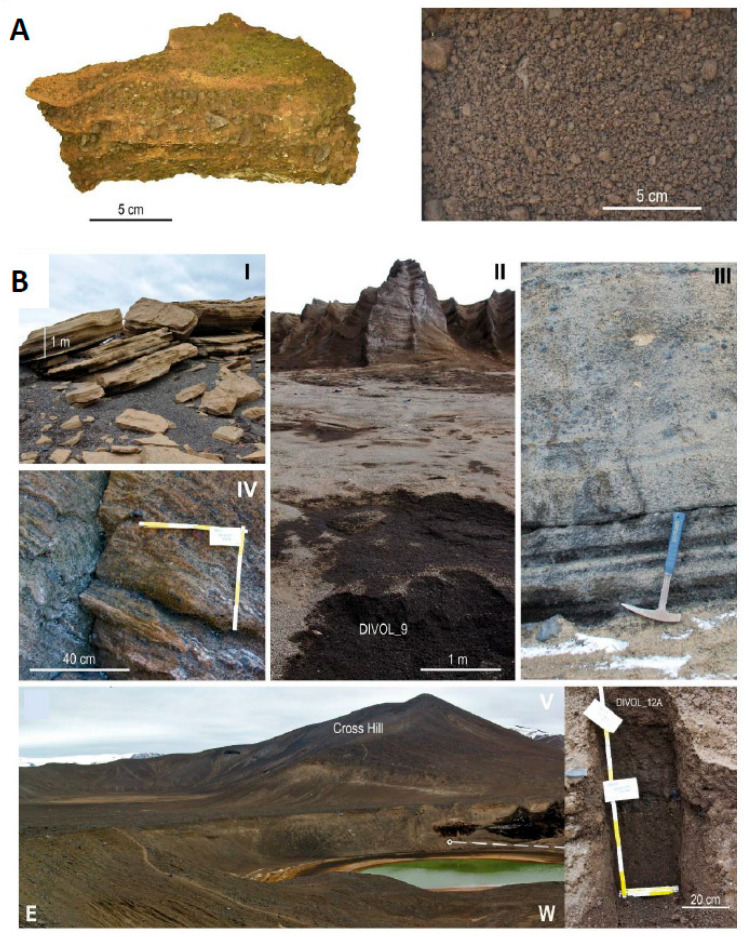
(**A**) Examples of samples. Image of samples DIVOL_23 (**left**) and DIVOL_12A (**right**) corresponding to a pyroclastic density current deposit and a loose pyroclastic lapilli-size fallout deposit, respectively. (**B**) Sampled outcrops. Images of some of the sampled outcrops (see Figure 1A for location and Table 1 for exact coordinates of the collected samples). I—Pyroclastic density current deposits located northeast of Wensleydale Beacon (Samples: DIVOL_4A and DIVOL_4D); II—Lapilli fallout deposits along the eastern flanks of Cross Hill volcanic edifice (Sample: DIVOL_9); III—Pyroclastic density current and lapilli fallout deposits at Irizar Lake (Samples: DIPV36_S1/S2/S3); IV—Pyroclastic density current deposits at Baily Head (Sample: DIVOL_23); and V—Lapilli fallout deposits along the northern flanks of Cross Hill volcanic edifice (Sample: DIVOL_12A).

**Figure 3 ijms-24-13824-f003:**
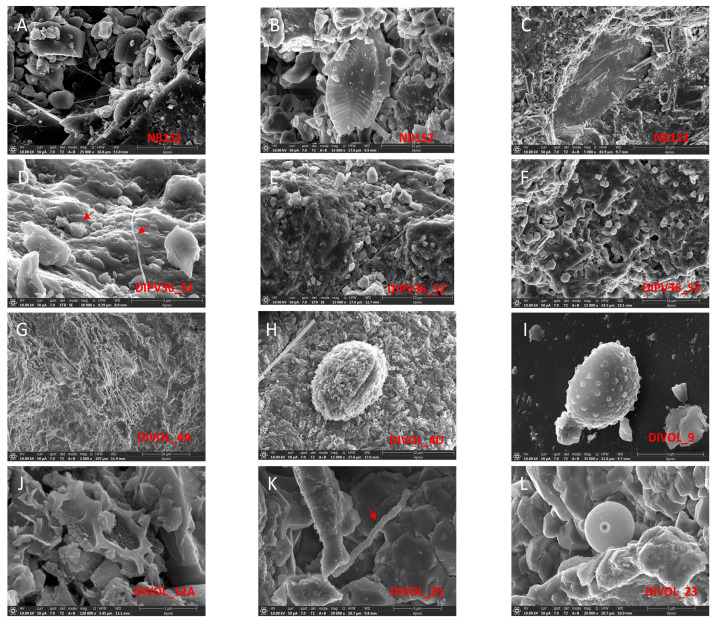
Scanning electron microscopy images of microorganisms in volcanic rock samples. (**A**) Filamentous fungus in sample NB1_S1. (**B**) Diatom frustule in NB1_S2. (**C**) Several diatom frustules in NB1_S3. (**D**) Group of cocci bacteria and filamentous fungus (indicated with red arrows) in sample DIPV36_S1. (**E**) Fungal hyphae in DIPV36_S2. (**F**) Coccus-shaped bacteria in DIPV36_S3. (**G**) Wood remains in DIVOL_4A. (**H**) Pollen grain in DIVOL_4D. (**I**) Fungus in DIVOL_9. (**J**) Diatom frustule fragment in DIVOL_12A. (**K**) Row of bacillus-shaped bacteria in DIVOL_23. (**L**) Chrysophyte cyst in DIVOL_23.

**Figure 4 ijms-24-13824-f004:**
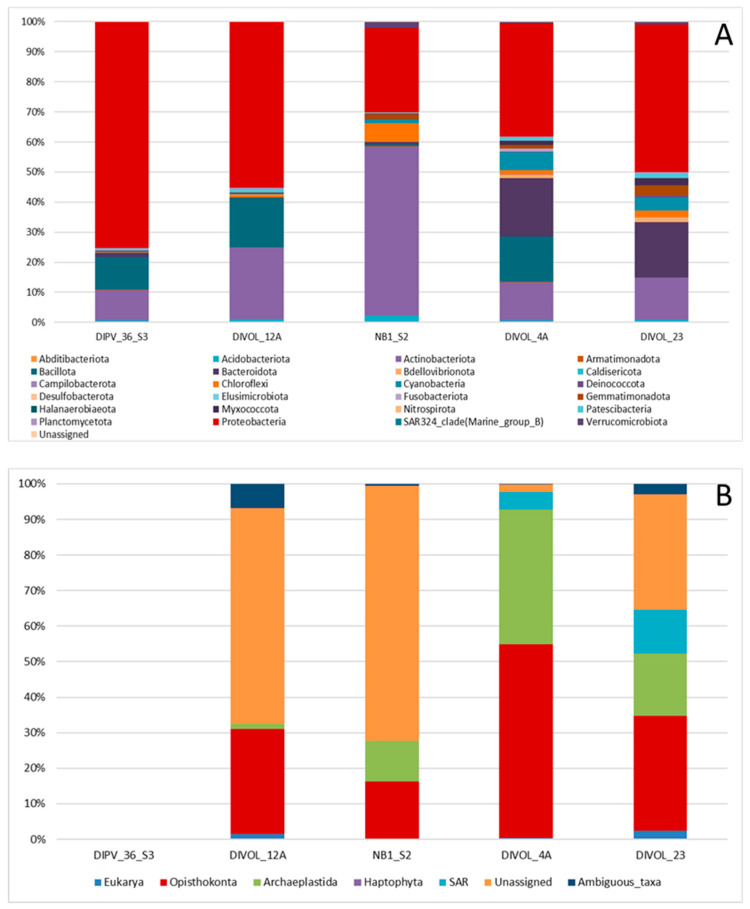
Microbial community distribution in the volcanic rock samples at the phylum level. Relative abundances of major taxa of bacteria and microeukaryotes based on (**A**) 16S rRNA and (**B**) 18S rRNA gene sequencing data, respectively.

**Figure 5 ijms-24-13824-f005:**
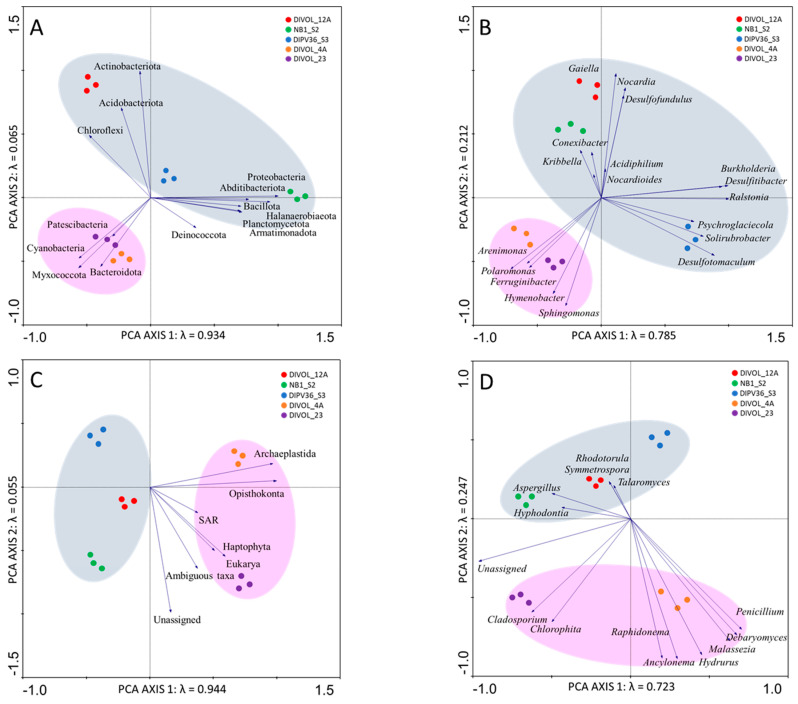
Principal component analysis (PCA). Scatterplot of (**A**) bacterial phyla, (**B**) bacterial genera, (**C**) eukaryotic phyla and (**D**) eukaryotic genera.

**Figure 6 ijms-24-13824-f006:**
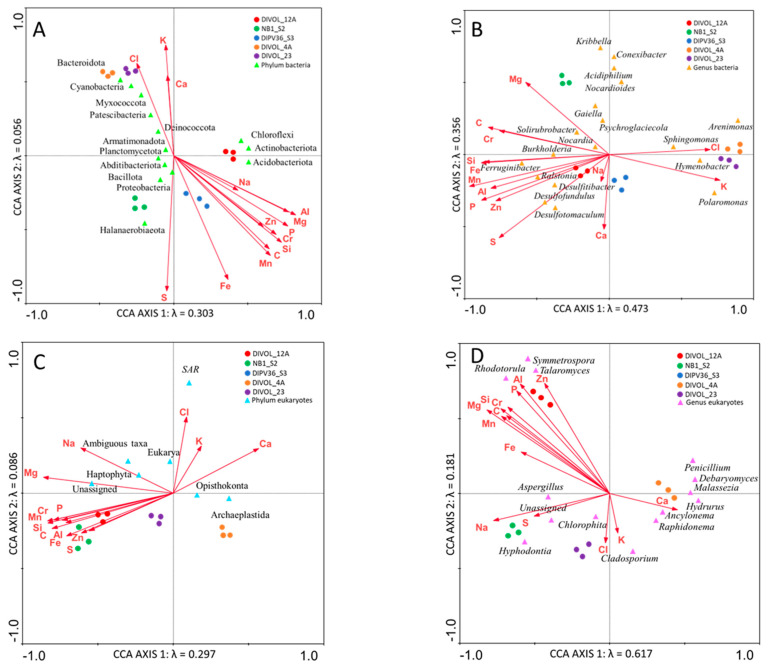
CCA of the microbial groups with regard to the chemical composition of samples. Analysis of (**A**) bacterial phyla, (**B**) bacterial genera, (**C**) eukaryotic phyla and (**D**) eukaryotic genera.

**Figure 7 ijms-24-13824-f007:**
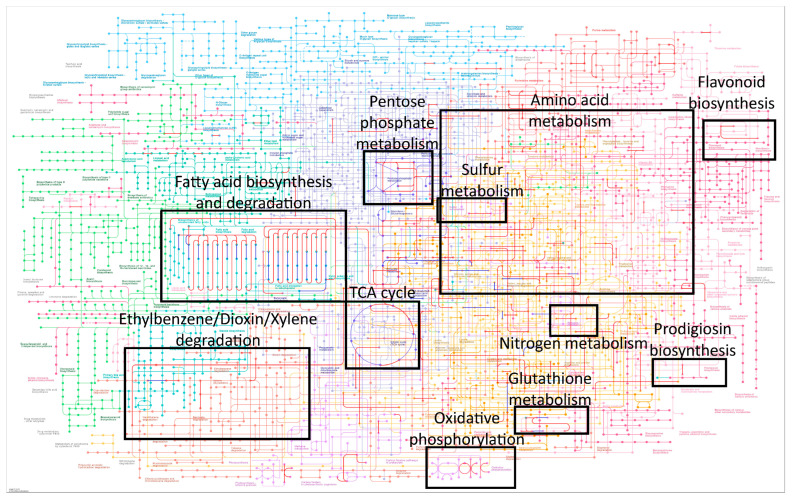
Metabolic pathways predicted from taxonomy. Metabolic pathways corresponding to the most abundant enzymes, inferred using the PICRUSt2 software (1% abundance, red; 5% abundance, blue).

**Table 1 ijms-24-13824-t001:** Name and location of the studied samples. International Generic Sample Number (IGSN) is also provided. Coordinates (x, y) are provided in UTM WGS84 20S coordinate system. (+, test performed −, test not performed).

Sample	IGSN	x	y	Rock Type	Picture	SEM	Metabarcoding
NB1_S1	IED18NB1S1	617,555.389	3,014,808.551	Pyroclastic rocks-loose lapilli	+	+	−
NB1_S2	IED18NB1S2	617,555.389	3,014,808.551	Pyroclastic rocks-loose lapilli	+	+	+
NB1_S3	IED18NB1S3	617,555.389	3,014,808.551	Pyroclastic rocks-loose lapilli	+	+	−
DIPV36_S1	IED18036S1	616,437.9272	3,014,413.246	Pyroclastic rocks-loose lapilli	+	+	−
DIPV36_S2	IED18036S2	616,437.9272	3,014,413.246	Pyroclastic rocks-loose lapilli	+	+	−
DIPV36_S3	IED18036S3	616,437.9272	3,014,413.246	Pyroclastic rocks-loose lapilli	+	+	+
DIVOL_4A	IED19004A	616,387	3,018,440	Pyroclastic density current deposit	+	+	+
DIVOL_4D	IED19004D	616,387	3,018,440	Pyroclastic density current deposit	+	+	−
DIVOL_9	IED190009	617,801	3,019,423	Pyroclastic rocks-loose lapilli	+	+	−
DIVOL_12A	IED19012A	617,509	302,0078	Pyroclastic rocks-loose lapilli	+	+	+
DIVOL_23	IED190023	626,739	3,015,998	Pyroclastic density current deposit	+	+	+

## Data Availability

Sequences obtained by 16S rRNA and 18S rRNA sequencing have been deposited in the NCBI Short Read Archive (SRA), BioProject PRJNA962314.

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
