# Peer review of "Adaptation of the Endolithic Biome in Antarctic Volcanic Rocks"

_ijms, 2023, doi:10.3390/ijms241813824_

Round 1
Reviewer 1 Report
Since this manuscript is titled "Adaptation of the endolithic biome in Antarctic volcanic rocks", I was looking forward to the "adaptation" aspect to be more discussed. I believe more compare and contrast with other comparable biome could elevate the quality of this manuscript.
Below are some specific comments/questions/suggestions:
Line 43: What is a "milometer"? I am sorry if I am not familiar with this term.
Line 331 : "axes" should be "axis"
Line 365 onwards: The reported metabolic pathways prediction seems generic. Is there anything "stand out" and specific to adaptation to Antarctic environment?
Line 426 onwards: What are the significance of producing these pigments? Any similar data reported from other non-antarctic endolithic biome?
If possible, I would like to see more comparison with other non-antarctic endolithic biome in the discussions. This would increase the interest towards the findings in this manuscript.
There was no further discussion about the endolith viability. What were the viable microorganisms and what can we learn from this?
Minor English corrections are required with some sentences.
Reviewer 2 Report
Hidalgo-Arias et al. present a work that could be defined as one of the best examples of a holistic approach to studying a defined problem, performed with a state-of-the-art methodology. They present both a compositional and functional analysis of some of the less studied microbiotas – that of the endolithic organisms. Additionally, this study is probably the unique of this type performed in Maritime Antarctica – a fact that gives additional weight to the importance of the results and the conclusions. These considerations motivate me to recommend this article for publication. However, some minor revisions should be done before accepting it:
Major remarks:
References are not organized according to the Journal’s requirements.
Minor remarks:
Line 43: “milometers”
Reviewer 3 Report
The manuscript by Hidalgo-Arias et all. presents the analysis of the endolithic biome in Antarctic volcanic rocks. These results are of potential interest to a broad audience, specifically those involved in environmental science and functional microbiome analysis in extreme environments. The paper is well-structured and written but it needs some improvements.
1. The 2.4 section name – I suggest changing the name to something like “analysis of the viability of endocytic microorganisms”. In my opinion, using the classical microbiology method for assessing viability had a big disadvantage – many of the environmental microbes are VBNC (viable but nonculturable). Please mention it in the discussion section. For further studies, I suggest improving the analysis with fluorimetric microscopy.
2. Please check the manuscript and italicize all the microorganisms' names.
3. Conclusion section – provide more information about the future directions of this research.
English language is ok, minor editing is required.
Reviewer 4 Report
The article “Adaptation of the endolithic biome in Antarctic Volcanic rocks.” Hidalgo-Arias et al. investigated the microbial consortium associated with two types of Antarctic volcanic rocks of Deception Island. They evaluated the chemical composition of the rock samples and employed electron microscopy to detect the presence of microbes inside the rock samples. Furthermore, Illumina sequencing was performed to identify the endolithic microbiota using 16S and 18S rRNA markers targeting the Prokaryotes and Eukaryotes.
PCA and CCA analyses were performed to detect 1) the relationship between microbial population and the rock types- pyroclastic density current deposits and loose pyroclastic lapilli-size deposits and 2) the relationship between microbial population and the chemical composition of the rock types, respectively. The PCA analysis segregated the microbial population according to the rock types, whereas the CCA analysis observed a close relationship between the microbes and ion composition.
The adaptation of these microbial assemblages was studied using PICRUSt tool -predicting the possible functional potential of observed microbes. The analyses revealed a most common functional relationship among the microbial communities and some adaptation mechanisms of microbes to defend themselves to survive in harsh environments.
The study indeed adds further information to the Antarctic microbial biodiversity. The article is well-articulated and easy to read.
Though the methods used here are technically sound, addressing the questions to prove the hypothesis, I wonder why authors used only 3 and 2 samples representing two different rock types for the high-throughput sequencing technique.
#Major comments
Lines 186-191 – Even though the references for 16S and 18S can inform the readers about the length, adding the sequencing read length of the target V3-V4 region in this section will be helpful.
Line 192 – How the mitochondria and chloroplast sequences were processed here? Adding some details will help the readers.
Line 258 – Did the study detect any archaea in the samples? If so, consider microbial ASVs instead of bacterial ASVs
Line 260 – could not be detected or could not be assigned?
Lines 324-326 – Missing the citations mentioned (Leps and Smilauer, 2003; Gloor et al., 2017) in the text within the reference section.
A proper justification for using these citations would help the readers here. Leps and Smilauer, 2003 describe the multivariate statistical analyses using CANOCO software packages. Gloor et al., 2017 discuss the pitfalls of treating the microbiome data and urge the researchers to treat the data as compositional.
Was this multivariate statistical analysis performed using CANOCO?
Please rewrite the sentence accordingly if the authors intend to say the strength of multivariate statistics in this study.
Line 332 – The values of the PCA axis are given in a Word document (SUPPLEMENTARY INFORMATION), not in a document with the file name Table S2. It isn't evident since another Excel document with Table S2 has taxonomic assignments, not PCA axis values.
Lines 390-393 – “The metabolism of microbial consortium” is vague. Consider revising to “The metabolic pathways and processes of the microbial consortium may be more complex…..”
Lines 447-470- The conclusion session in the article is very general and not based on the studies performed here. Please rewrite the conclusion focusing on the relevant findings relating it to astrobiology.
